# Antifungal Testing of Vaginal *Candida* Isolates in Pregnant Women: A Retrospective, Single-Center Study in Adana, Türkiye

**DOI:** 10.3390/jof11020092

**Published:** 2025-01-24

**Authors:** Mete Sucu, Nevzat Ünal, Ayşe Sultan Karakoyun, İrem Şahin, Oğuzhan Bingöl, Fatih Hüner, Fatma İşlek Uzay, İlker Ünal, Dilek Yeşim Metin, Macit Ilkit

**Affiliations:** 1Division of Perinatology, Department of Obstetrics and Gynecology, Faculty of Medicine, Çukurova University, Adana 01330, Türkiye; metesucu@yahoo.com (M.S.); 91f.islek@gmail.com (F.İ.U.); 2Division of Mycology, Department of Microbiology, Faculty of Medicine, Çukurova University, Adana 01330, Türkiye; drnevzatunal@gmail.com (N.Ü.); md.aysesultank@gmail.com (A.S.K.); oguzhan-bingol@hotmail.com (O.B.); fatihhunerr@gmail.com (F.H.); 3Laboratory of Medical Microbiology, Adana City Training and Research Hospital, University of Health Sciences, Adana 01370, Türkiye; 4Division of Mycology, Faculty of Medicine, University of Ege, Bornova, Izmir 35100, Türkiye; irem_nur_sahin@hotmail.com (İ.Ş.); dilekyesimb@yahoo.com (D.Y.M.); 5Department of Biostatistics, Faculty of Medicine, Çukurova University, Adana 01330, Türkiye; ilkerun@cu.edu.tr

**Keywords:** antifungal susceptibility tests, *Candida albicans*, *Candida glabrata*, fluconazole, petite, vaginitis

## Abstract

Clinical and mycological data are essential for the optimal management of patients with *Candida* vaginitis (CV), particularly in cases of (i) azole-resistant *C*. *albicans* vaginitis, (ii) recurrent CV, and (iii) CV in pregnant women. The present retrospective single-center study investigated the antifungal activity of six commonly used antifungals against randomly selected vaginal isolates recovered from 68 pregnant women in Adana, Türkiye, including *C*. *albicans*, petite *C*. *glabrata*, non-petite *C*. *glabrata*, and *C*. *krusei*, using the disk diffusion method at pH 4 and 7. Furthermore, the antifungal activities of fluconazole and itraconazole were also assessed using the broth microdilution method. For all isolates, the mean inhibition zone diameters were narrower for itraconazole and ketoconazole and larger for miconazole at pH 4 than pH 7 (*p* < 0.05). For nystatin, zone diameters were wider in *C*. *albicans* and petite *C*. *glabrata* at pH 4 (*p* < 0.001 and *p* < 0.001). Remarkably, clotrimazole was more active at pH 4 than at pH 7, except against non-petite *C*. *glabrata* isolates. Based on the broth microdilution results, the resistance rate was higher at pH 4 than at pH 7 in all isolates. *Candida glabrata* petite isolates exhibited MIC values 2 to 5 times higher than those of the non-petite isolates for both fluconazole and itraconazole. This study highlights the potent activity of topical antifungals (miconazole, nystatin, and clotrimazole) for the treatment of CV in pregnant women and highlights the need to identify petite and non-petite mutants of vaginal *C*. *glabrata* isolates to obtain more reliable data and for antifungal susceptibility testing prior to decision-making. The results of the two antifungal susceptibility methods were compared for *C. albicans* and *C. glabrata* isolates, and the reliability of the disk diffusion test was discussed.

## 1. Introduction

*Candida* vaginitis (CV) is one of the most common mycoses globally. Eight out of ten women experience a CV attack, and 40–45% experience two or more episodes [1,2,3]. However, the epidemiological data required to make realistic and sustained estimates of CV are limited [3,4]. CV is almost always treated with fluconazole or topical azole antifungals with fungistatic activity against *Candida* species. Despite their widespread use in the treatment of acute CV (ACV)/recurrent CV (RCV), azoles are inadequate for treating infections caused by non-*albicans Candida* species and azole-resistant *C*. *albicans*, and their effect decreases at low pH (4–4.5) [2,5,6]. *Candida glabrata*, the most common cause of non-*albicans Candida* vaginitis, is resistant to azoles and is on the rise globally; therefore, treatment is challenging, with limited therapeutic options [7,8,9,10]. Although *C*. *albicans* remains a common causative agent, the significant decrease in its prevalence is alarming because it can be treated more easily than non-*albicans Candida* vaginitis. Additionally, the dramatic increase in *C*. *glabrata* and *C*. *krusei* prevalence causes azole-resistant CV [1,7,11]. The reasons for this increase in non-*C*. *albicans* species are thought to include (i) the unnecessary use of azole antifungals, (ii) the widespread use of single-dose treatment options, (iii) low-dose azole maintenance therapy, and (iv) poor adherence of patients to the proper therapeutic regimen [2,7,11]. Overall, there is a shortage of effective oral and topical antifungal compounds for treating CV, particularly in pregnant patients in whom the use of oral antifungals is contraindicated [1,4,7,11].

The current Clinical & Laboratory Standards Institute (CLSI) and European Committee on Antimicrobial Susceptibility Testing (EUCAST) guidelines do not recommend the routine testing of vaginal isolates. Therefore, in addition to the local antifungal resistance profile, it is important to identify multiple-azole resistant isolates and antifungal drugs that are active at vaginal pH (4–4.5) to effectively manage treatment-resistant and recurrent CV [12]. This is gaining importance in difficult-to-treat CV cases, where resistance to topical antifungals is common [1,2,3,7,11]. To overcome this problem, Dr Sobel’s group [5,13,14,15,16,17] proposed antifungal testing at pH 4.5 using the CLSI broth microdilution method, which simulates the vaginal pH. These leading studies raised awareness for testing vaginal *Candida* isolates in treatment-resistant cases and RCV and revealed that fluconazole resistance is not as low as expected [15,17]. Hence, antifungal susceptibility testing (AFST) is required to identify a reliable drug-of-choice and to clinically manage patients [5,7,13]. Importantly, CV has limited therapeutic options during pregnancy, and only topical antifungals are recommended. Notably, ketoconazole is no longer recommended for treating CV [18]. Additionally, oral azoles should be avoided in women who are (i) pregnant, (ii) planning pregnancy, or (iii) breastfeeding [18].

Small and slow-growing colonies of *C*. *glabrata* “petite variants” are often not considered for AFST by microbiology laboratories. Isolating and identifying such phenotypes is extremely important during treatment, and continuing azole therapy will favor the selection of the pathogen population, which can result in therapeutic failure and poor clinical outcomes [19,20,21]. Consistent with this, we have previously reported that a notable number of clinical *C*. *glabrata* isolates are in fact a mixture of both large and petite colonies [20]. Notably, petite isolates also have a high tolerance to echinocandins [20,22]. Therefore, properly characterizing and reporting petite colonies of *C*. *glabrata* in clinical samples holds immense clinical importance. A recent study from our hospital revealed no “petite variants” among 27 *C*. *glabrata* vaginitis isolates, suggesting that its prevalence varies even within the same hospital [23].

These findings motivated us to explore the presence of *C*. *glabrata* “petite variants” in our vaginal stock collection from pregnant women and to compare the antifungal testing results with those of their parent isolates. The available *C*. *albicans* and non-*C*. *albicans* strains were also investigated using the disk diffusion and broth microdilution methods to reveal epidemiological data in Adana, Türkiye. Specifically, both tests were performed at pH 4 and 7, and the results were compared. Additionally, the suitability and reliability of the disk diffusion method were assessed. The findings of the present study could enhance our understanding of the antifungal susceptibility profiles of “petite variants” of *C*. *glabrata* in vaginal isolates.

## 2. Materials and Methods

### 2.1. Study Group and Identification of Isolates

The population studied and selection of isolates are detailed in Figure 1. Vaginal swab samples were cultured on CHROMagar *Candida* plates (CHROMagar, Paris, France) [24] and diagnoses of CV were established based on a combination of clinical and mycological findings [25]. All isolates were identified to the species level using MALDI-TOF MS (Bruker Daltonics, Bremen, Germany), the Bruker Biotyper 3.1 software and its database (version 3.1.66) [26] and stored at −20 °C.

Subsequently, the isolates were plated onto Sabouraud glucose agar (SGA) plates (Merck, Darmstadt, Germany) for use in the study, and the plates were incubated for 48 h at 37 °C. Only revivable and pure isolates were included in the study.

### 2.2. Candida glabrata “Petite Variants”

*Candida glabrata* colonies were sub-cultured onto SGA plates using the single-colony streaking technique. The plates were incubated at 36 °C for 48–72 h, after which colony morphology was evaluated daily. If normal-sized (parent) and small pinpoint colonies grew on the same SGA plate, the colonies were individually subcultured onto SGA [21]. For the final identification, *C*. *glabrata* colonies were subcultured on yeast–peptone–glycerol (YPG) agar supplemented with 10 g/L yeast extract (Merck), 20 g/L peptone (Merck), and 20 g/L glycerol (Isolab, İstanbul, Türkiye) and incubated at 36 °C for 72 h. Parent colonies were grown on YPG agar; however, small pinpoint colonies that could not grow on YPG (because these colonies could not ferment glycerol) were considered “petite mutants” [20].

### 2.3. Antifungal Testing

#### 2.3.1. Disk Diffusion Method

Antifungal susceptibility tests were performed using the disk diffusion method, based on the CLSI M44-A2 document. For this, Mueller–Hinton agar (Merck) containing 2% glucose (Merck) and 0.5 µg/mL methylene blue (Merck) was prepared [27]. The pH of the prepared medium was adjusted to 4 or 7 using NaOH (Isolab) or HCl (Isolab), as required [28]. For each isolate, a 0.5 McFarland turbidity yeast suspension was prepared in approximately 5 mL of sterile saline using 24 h fresh cultures passaged on SGA. Yeast suspensions were spread homogeneously on two separate media plates prepared at pH 4 and 7. Fluconazole (FLU; 25 μg, Bioanalyse, Ankara, Türkiye), itraconazole (ITR; 10 μg, Bioanalyse), ketoconazole (KTC; 50 μg, Bioanalyse), clotrimazole (CLT; 50 μg, Bioanalyse), miconazole (MCZ; 50 μg, Bioanalyse), and nystatin (NY; 100 U, Bioanalyse) disks were added to the test media. The diameters of the zones of inhibition determined at the end of the incubation period (24–48 h at 36 °C) were recorded.

The resistance cut-off values for fluconazole were evaluated as susceptible (S), dose-dependently susceptible (S-DD), and resistant (R) according to CLSI M44-A2 [27]. Other antifungal drugs were reported according to the values presented in Appendix A, which were adapted from the literature as resistance cut-off values have not yet been reported [29,30]. The quality control strains *C*. *krusei* ATCC 6258 and *C*. *parapsilosis* ATCC 22019 were included in all assays. All tests were performed in duplicate.

#### 2.3.2. Broth Microdiution Method

Susceptibility testing for fluconazole and itraconazole was conducted on viable isolates using standard broth microdilution methods, following the CLSI M27-A3 and CLSI M60 guidelines [31,32]. The tests were performed in RPMI 1640 medium (Merck) buffered with MOPS (Merck) and supplemented with 0.2% glucose, and the cultures were incubated at 35 °C for 24 and 48 h. The pH of the prepared medium was adjusted to 4 or 7 using NaOH or HCl, as in the disk diffusion method [28]. Fluconazole resistance breakpoints were evaluated according to the CLSI M27-A3 and M27-S4 guidelines [31,33]. Since the CLSI does not provide specific resistance breakpoints for itraconazole, epidemiological cut-off values were utilized [31,34] (Appendix A).

### 2.4. Statistical Analysis

Categorical measurements are summarized as numbers and percentages, and numerical measurements are summarized as the mean and standard deviation. The sensitivity or resistance of microorganisms to antifungals and their compatibility or incompatibility at different pH values (4 and 7) were examined based on McNemar’s test. Whether the inhibition zone diameters met the assumption of a normal distribution was tested with the Shapiro–Wilk test. A *t*-test of dependent groups was used to compare the inhibition diameters at pH 4 and pH 7. One-way analysis of variance was used for a general comparison of inhibition diameters according to microorganisms. Games-Howell tests were used in pairwise comparisons of groups for situations found to be significant in this analysis. To assess the relationships between the inhibition zone diameters at pH 4 and pH 7 for all isolates, Pearson’s correlation coefficient was used. To control the influence of microorganisms on the correlations between the inhibition zone diameters at pH 4 and 7, a partial correlation analysis was employed. IBM SPSS Statistics 20 (IBM Corp., Armonk, NY, USA) was used to statistically analyze the data. The statistical significance level was set at 0.05 for all tests.

### 2.5. Ethics Statement

This study was conducted in accordance with the local legislation and institutional requirements. The experimental protocols were approved by the Ethics Committee of Çukurova University Faculty of Medicine after thorough scrutiny of the proposed procedures (no. 14.02.2020/87).

## 3. Results

The performance of the reference strains evaluated with the two AFST methods was within the expected range. The susceptibility profiles of *C*. *albicans* (*n* = 29), petite *C*. *glabrata* (*n* = 26), non-petite *C*. *glabrata* (*n* = 26), and *C*. *krusei* (*n* = 13) isolates to six antifungal drugs (fluconazole, itraconazole, clotrimazole, ketoconazole, miconazole, and nystatin) at pH 7 and pH 4 are presented in Figure 2**,** and the mean inhibition zone diameter values are presented in Appendix A. Appendix A presents a correlation analysis between the variables (pH 4 and pH 7) to show the consistency of each isolate’s pH changes within the species. For all isolates, the mean inhibition zone diameters were narrower for itraconazole and ketoconazole and larger for miconazole at pH 4 than at pH 7 (*p* < 0.05). The mean inhibition zone diameter of fluconazole decreased at pH 4 compared with that at pH 7 (*C*. *albicans*, petite *C. glabrata*, and non-petite *C. glabrata*, *p* < 0.001, *p* = 0.039, and *p* = 0.01, respectively), except for *C*. *krusei* (*p* < 0.01), whereas the average inhibition zone diameter of clotrimazole increased at pH 4 compared with that at pH 7 (*C*. *albicans*, petite *C*. *glabrata* and *C*. *krusei*, *p* = 0.037, *p* < 0.001, and *p* < 0.001, respectively), except for non-petite *C*. *glabrata* (*p* = 0.277). The broth microdilution susceptibility results and MIC values for fluconazole and itraconazole are shown in Appendix A and Table 1. According to the broth microdilution test performed at pH 4, the resistance rate was higher compared to that at pH 7 in all isolates. Furthermore, while evaluating the broth microdilution tests, we observed that MIC determination at pH 4 was clearer and easier to interpret.

*Candida albicans* isolates (*n* = 29) demonstrated a significant correlation in susceptibility to fluconazole, itraconazole, and ketoconazole between pH 4 and pH 7 (*p* < 0.01). Although all *C*. *albicans* isolates were susceptible to miconazole and clotrimazole at pH 4, their susceptibilities to miconazole and clotrimazole were lower at pH 7 (24 isolates for miconazole and 26 for clotrimazole) (*p* > 0.05). Furthermore, all *C*. *albicans* isolates were susceptible to nystatin at both pH 7 and 4, with larger mean inhibition zone diameters at pH 4 (*p* < 0.001). In *C. albicans* isolates, both disk diffusion and broth microdilution test results were consistent for fluconazole, whereas inconsistent results were observed for itraconazole. Furthermore, in *C. albicans* isolates, the minimum inhibitory concentration (MIC) ranges for fluconazole at pH 4 were similar to those at pH 7, whereas the MIC ranges for itraconazole were narrower; the MIC values inhibiting the growth of 90% of isolates (MIC_90_) for both drugs were found to be two dilutions lower. The antifungal susceptibility profiles of petite (*n* = 26) and non-petite (*n* = 26) *C*. *glabrata* isolates are presented in Figure 2. In petite *C*. *glabrata* isolates, susceptibility results for fluconazole and itraconazole exhibited a significant correlation between pH 4 and pH 7, whereas in non-petite *C*. *glabrata* isolates, the correlation between itraconazole and nystatin was significant (*p* < 0.01). Although all *C*. *glabrata* colonies were susceptible to miconazole at pH 4, the mean inhibition zone diameter was greater at pH 4 than at pH 7 (*p* < 0.001, Figure 3). Clotrimazole was more effective against petite *C*. *glabrata* isolates at pH 4 (96.2%, 25/26) than at pH 7 (38.5%, 10/26) (*p* < 0.05), whereas it was only effective against 42.3% (11/26) and 50% (13/26) of non-petite *C*. *glabrata* isolates, respectively (*p* > 0.05). All isolates were sensitive to nystatin at both pH values. Furthermore, for all tested antifungals, the mean inhibition zone diameters were greater in petite colonies than those in non-petite colonies at pH 4. Consistent results were observed in both the disk diffusion and broth microdilution tests for fluconazole in petite and non-petite *C. glabrata* strains; the itraconazole results agreed with the disk methods results at pH 7, while the pH 4 results were inconsistent with these test outcomes. Additionally, petite strains of *C*. *glabrata* exhibited MIC values 2 to 5 times higher than those of the non-petite strains for both fluconazole and itraconazole. Furthermore, the MIC ranges observed for both petite and non-petite *C*. *glabrata* at pH 4 were at least three dilutions lower than those at pH 7, and the MIC_90_ values were at least one dilution higher.

Although all *C*. *krusei* isolates (*n* = 13) were resistant to fluconazole at pH 7, only 69.2% (9/13) were resistant to fluconazole at pH 4. The susceptibility to miconazole was 30.8% (4/13) at pH 7, and all isolates (100%, 13/13) were susceptible at pH 4 (*p* < 0.05). All *C*. *krusei* isolates were susceptible to clotrimazole (100%, 13/13) at both pH 4 and 7; however, the mean inhibition zone diameters were larger at pH 4 (*p* < 0.001; Table 1). The disk diffusion and broth microdilution susceptibility test results were inconsistent for itraconazole (Appendix A).

## 4. Discussion

The antifungal susceptibility profiles of vaginal *Candida* isolates to six antifungal compounds were tested, and the available options for treating CV during pregnancy were highlighted. Additionally, petite and non-petite *C*. *glabrata* isolates derived from the same patients were further examined and compared. The results indicated that resistance to topical antifungal compounds was uncommon. Miconazole and nystatin were specifically more active against *C*. *albicans* and *C*. *glabrata* isolates at pH 4 than at pH 7. Notably, topical nystatin is not commercially available in Türkiye. In the present study, 51% of all isolates (48/94) were susceptible to fluconazole at pH 7, whereas 29% (27/94) were susceptible to fluconazole at pH 4 (*p* < 0.05). Only 29% (27/94) and 20% (19/94) of the isolates were susceptible to itraconazole at pH 7 and 4, respectively (*p* > 0.05). Therefore, mycological information is required to reliably treat CV in pregnant women. The present study also revealed that miconazole and clotrimazole are the best therapeutic options for CV in pregnant women in Adana, Türkiye.

An ARTEMIS survey indicated that when using invasive infection isolates, the fluconazole disk diffusion test is especially reliable for detecting resistant *Candida* species isolates through reference MIC testing, which has been performed with increasing accuracy worldwide [35]. Importantly, the level of agreement between disk diffusion and MIC results in these studies was the highest for *C*. *albicans* and lowest for *C*. *glabrata*. The latter was almost entirely due to major errors (resistance in disk diffusion and susceptibility in the broth microdilution test) [35,36]. Our study showed a high level of agreement for *C*. *albicans* and *C*. *glabrata* isolates, but significant discrepancies in itraconazole susceptibility were noted for petite *C*. *glabrata* and *C*. *krusei* isolates. Given the limited number of isolates, these findings should be interpreted with caution and require further validation with a larger sample size to rule out potential random errors. The disk diffusion method has been compared with the broth microdilution method, and several advantages (easier, inexpensive, and easy-to-interpret) and disadvantages (does not provide MICs) have been noted [35,36].

Sobel and Akins [15] tested 125 vaginal *C*. *albicans* isolates against fluconazole. At pH 7, 42.3% (*n* = 54) were initially considered susceptible to fluconazole; however, at pH 4.5, 31% (17/54) of the isolates were considered resistant, and tests performed at an acidic pH were more reliable. Notably, fluconazole susceptibility in vaginal *C*. *albicans* isolates longitudinally obtained from women with RCV remained predominantly stable despite azole avoidance, and only 5.2–10.5% of the susceptible isolates become resistant or vice versa [16]. A 10-year U.S. study noted that *C*. *albicans* constituted at least 71.8% of CV isolates, and 44 out of 193 (22.8%) isolates were reported to be fluconazole-resistant (MIC ≥ 8 µg/mL). Furthermore, 52% and 38% of the *C*. *albicans* isolates exhibited susceptibility at pH 7 and 4.5, respectively [17]. In our previous study at the same hospital, 207 vaginal isolates recovered from pregnant women were tested against 13 antifungal drugs using the CLSI broth microdilution method (M27-A3). Clotrimazole, ketoconazole, and miconazole were 90–98% active against the strains. Briefly, 1.9% of the isolates were fluconazole-resistant, and 6.8% exhibited S-DD [37]. In the present study, the susceptibility profiles to fluconazole, itraconazole, and ketoconazole were lower at pH 4 than pH 7.

Boikov et al. [13] tested 108 vaginal isolates—60 *C*. *albicans* and 48 non-*albicans Candida* strains—against a new echinocandin, CD101, and found that it has potent activity at both pH 4 and 7. Many fluconazole-resistant isolates, particularly *C*. *glabrata*, are resistant to itraconazole. In another study, ibrexafungerp was found to be effective against 187 vaginal *Candida* isolates [fluconazole-resistant (*n* = 52), fluconazole-susceptible *C*. *albicans* (*n* = 30), and non-*albicans Candida* (*n* = 100)] at pH 4 and 7, and the MIC did not change with pH [14].

Notably, antifungal susceptibility tests have not been standardized or validated for non-*albicans Candida* species. Therefore, empiric therapy is recommended for at least 30–40% of CV cases [11]. Recently, Dunaiski et al. [9] contributed to our in-depth-understanding of why *C*. *glabrata* vaginitis causes treatment-resistant infections using whole-genome sequencing, identifying a clone that is predominant in women with *C*. *glabrata* vaginitis and harbors various mutations in resistance-associated genes. Interestingly, in our study, the susceptibility profiles of petite and non-petite *C*. *glabrata* isolates potentially varied at different pH values, and clotrimazole was more effective against petite isolates than non-petite (parent) isolates recovered from the same patient. This finding contradicts earlier findings reporting that petite isolates are more resistant than non-petite isolates, as we observed that petite strains exhibited higher MIC values for fluconazole and itraconazole than non-petite strains [20,21]. We suggest that the susceptibility profiles of mitochondrial mutants in *C*. *glabrata* remain poorly understood. However, according to our stock records, 51.2% of vaginal *C*. *glabrata* isolates are “petite variants”, warranting a closer investigation of “petite variants” associated with recurrent and recalcitrant infections, and poor prognosis. The present study and recent work by our group [23] revealed that the prevalence of “petite variants” in vaginal isolates varies from 0% to 51.2%; thus, determining “petite variants” and their clinical relevance is important.

AFST is crucial for effective treatment, but its access is limited in resource-constrained regions such as Africa, the Asia–Pacific, and Latin America, like many laboratories in our country [38]. Various methods of detecting antifungal susceptibility profiles are available, including phenotypic, molecular, and automated systems. Once their validity is confirmed, cost-effective methods that do not require specialized equipment or extensive experience are preferred [38]. Broth microdilution methods are reference methods in the CLSI and EUCAST guidelines, but they are not for day-to-day use and not readily accessible due to the high costs associated with certain test guidelines, such as those of the CLSI, as well as significant equipment requirements [31,38]. Simple disk tests provide rapid and reliable results for many common *Candida* species [35,36]. However, owing to the limited sample size and potential discrepancies for certain isolates, the reliability of the disk diffusion method should be interpreted with caution, as noted in the present study. Moreover, easily accessible media for this method, such as Mueller–Hinton agar, reduce the cost of the method and ensure reproducibility [27,38]. Two of the timeframe problems with AFST are (i) a need to know the fungal species before an interpretation can be issued (for the CLSI and/or EUCAST) and (ii) isolate transport and regrowing delays in obtaining the answer, as observed in our study. The present study had several limitations, primarily (i) a limited number of study isolates, which represents a significant constraint on the generalizability of the findings and impedes the comparability of the methods. Additionally (ii) its retrospective nature and (iii) data are limited to in vitro test findings, and data on clinical treatment and outcomes are lacking. Further studies can explore the steps necessary to address the limitations of the current study.

## 5. Conclusions

This study highlights the importance of examining susceptibility differences between petite and non-petite *C*. *glabrata* variants, as well as comparing them with other *Candida* species, in managing CV. These findings are particularly crucial for pregnant patients who cannot use oral antifungals, emphasizing the need for alternative treatment strategies. Overall, AFST is vital for patient care guidance, and this will prevent inappropriate therapy and lead to rapid recovery from CV. In addition, to assess the size of the resistance problem at the regional or national level, understanding the current profile is encouraged. Importantly, disk diffusion is an applicable method for improving access to resource-constrained laboratories. Therefore, in addition to the local antifungal resistance profile, it is important to identify azole-resistant isolates and antifungal drugs that are active at vaginal pH (4–4.5). Owing to the high prevalence of CV and its severe morbidity, clinicians and clinical mycologists should work collaboratively to manage this global disease.

## Figures and Tables

**Figure 1 jof-11-00092-f001:**
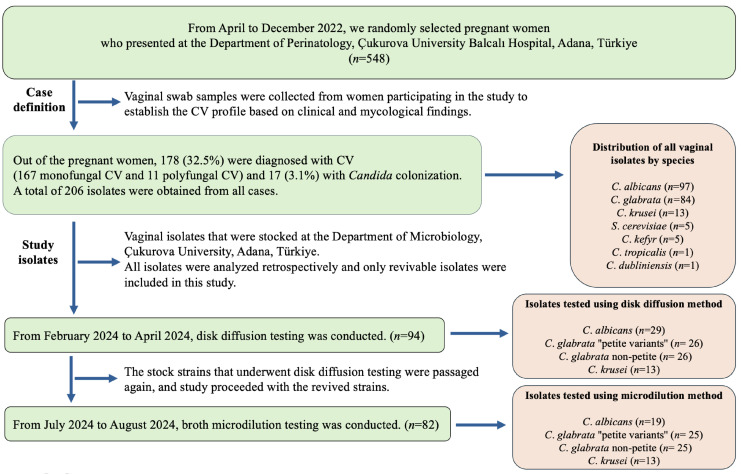
Flow chart presenting the population studied and selection of isolates.

**Figure 2 jof-11-00092-f002:**
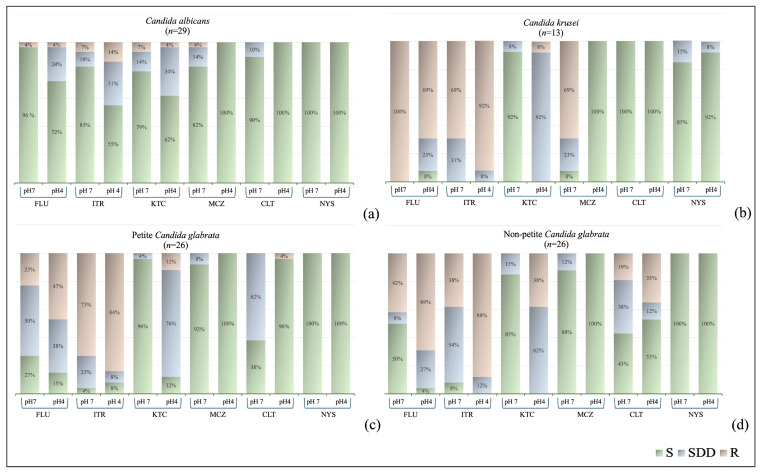
Disk diffusion method results of vaginal *Candida* isolates at pH 4 and pH 7: (**a**) *Candida albicans*, (**b**) *Candida krusei*, (**c**) petite *Candida glabrata*, (**d**) non-petite *Candida glabrata*. S, Susceptible; SDD, Susceptible-dose-dependent; R, Resistant. FLU, Fluconazole; ITR, Itraconazole, CLT; Clotrimazole, KTC, Ketoconazole; MCZ, Miconazole; NY, Nystatin.

**Figure 3 jof-11-00092-f003:**
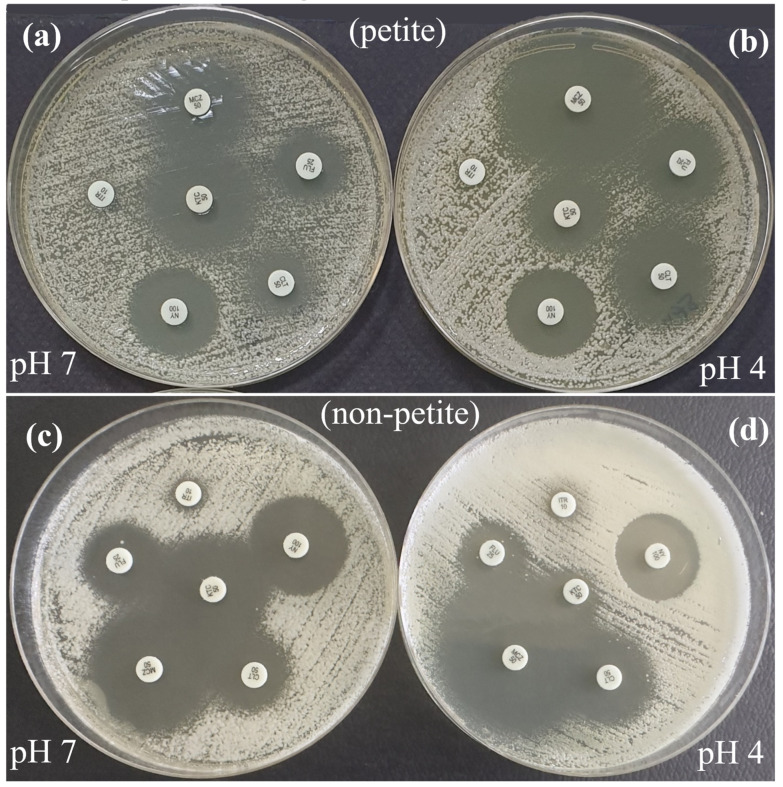
Antifungal susceptibility zone diameters of *C*. *glabrata* petite (**a**,**b**) and non-petite (**c**,**d**) colonies on Mueller–Hinton agar, pH 7 (**a**,**c**) and pH 4 (**b**,**d**), using the disk diffusion method. In petite colonies, the zone diameters for KTC at pH 7 and MCZ, CLT, and FLU at pH 4 were larger. In non-petite colonies, the zone diameters for KTC, FLU, and NY were larger at pH 7, while MCZ and CLT were larger at pH 4. FLU, Fluconazole; ITR, Itraconazole, CLT; Clotrimazole, KTC, Ketoconazole; MCZ, Miconazole; NY, Nystatin.

**Table 1 jof-11-00092-t001:** In vitro antifungal susceptibility metrics of vaginal *Candida* isolates, using CLSI M27-A3 standard.

		Fluconazole				Itraconazole			
		MIC Range	MIC_50_	MIC_90_	GM	MIC Range	MIC_50_	MIC_90_	GM
pH 7	*C. albicans* (*n* = 19)	<0.125–4	2	4	0.89	<0.03–2	0.25	2	0.18
	Petite *C. glabrata* (*n* = 25)	2–>64	16	64	16.95	0.25–>16	1	16	1.63
	Non-petite *C. glabrata* (*n* = 25)	0.5–>64	8	16	6.41	0.03–2	0.5	1	0.50
	*C. krusei* (*n* = 13)	16–>64	32	32	30.33	0.25–1	0.5	0.5	0.42
pH 4	*C. albicans* (*n *= 19)	<0.125–8	0.5	1	0.60	0.125–0.5	0.25	0.25	0.26
	Petite *C. glabrata* (*n* = 25)	8–>64	>64	>64	49.87	2–>16	16	>16	10.85
	Non-petite *C. glabrata* (*n* = 25)	8–>64	64	64	39.95	1–>16	8	16	7.57
	*C. krusei* (*n* = 13)	16–>64	32	32	28.76	0.5–1	1	1	0.85

MIC, minimum inhibitory concentration; GM, geometric mean.

## Data Availability

The original contributions presented in this study are included in the article/Appendix A. Further inquiries can be directed to the corresponding author.

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
