# Peer review of "Antifungal Testing of Vaginal Candida Isolates in Pregnant Women: A Retrospective, Single-Center Study in Adana, Türkiye"

_jof, 2025, doi:10.3390/jof11020092_

Round 1

Reviewer 1 Report (Previous Reviewer 1)

1. The topic and aims of this manuscript are not important
2. The results of this manuscript provided limited new information to the readers
3. The case number of the study is inadequate to conclude the results

1. The topic and aims of this manuscript are not important
2. The results of this manuscript provided limited new information to the readers
3. The case number of the study is inadequate to conclude the results

Author Response

1.The topic and aims of this manuscript are not important.

Response: We consider this study important because of the limited data on the prevalence and antifungal testing of vaginal petite and non-petite Candida glabrata isolates. The study also examined antifungal susceptibility (both pH 4 and pH 7) via the low-cost disk diffusion method. These findings could improve the management of vaginal Candida infections, owhich are often overlooked in antifungal management strategies.

2.The results of this manuscript provided limited new information to the readers.

Response: We appreciate the feedback. While the findings may seem limited, we believe that this study provides valuable insights into antifungal responses and the effect of vaginal pH and contributes to filling the gap in the current literature on Candida vaginitis.

3.The case number of the study is inadequate to conclude the results
Response: We acknowledge that we were able to access a limited number of patients and isolates, and we agree that larger-scale studies are needed on this topic. We hope that these findings will serve as a foundation for future research.

Reviewer 2 Report (Previous Reviewer 3)

The investigation of potential differences in susceptibility to topical antifungals between petite and non-petite variants of C. glabrata, as well as comparisons with other Candida species, is crucial for improving the management of vaginitis caused by these pathogens, particularly in pregnant patients for whom oral antifungals are not an option. 

It is also essential to conduct these tests under conditions that mimic vaginal pH to assess whether pH influences antifungal efficacy. For meaningful insights, the results must be comparable to findings from other research groups.

The authors compared the disk diffusion method with the microdilution method, which is considered the reference method. However, they did not statistically analyze or extensively discuss the significant differences between the results obtained by both methods. While there are no established reference values for all antifungals in the microdilution method used in the study, variations exist even among the antifungals common to both methods. The reliability of the disk diffusion method for the analyzed samples is not fully validated based on the results presented.

Lines 19 -20: The sentence is misconstrued.

Line 210: It should be "microorganisms"

Author Response

Reviewer 2:

(a). The authors state (lines 38-39) that the results from the two antifungal susceptibility methods are consistent for C. albicans and C. glabrata isolates. However, they do not describe the statistical analysis used to compare the results obtained from both methods.

Response: As indicated in the study limitations section of the manuscript, the sample size of the study is limited, and the limited sample size makes the results of antifungal susceptibility methods among microorganisms incomparable. In particular, the results of the analyses were inconclusive. Consequently, the findings are presented through numerical comparisons. If the study is replicated with a larger sample size while maintaining the rates obtained in the original study, it becomes evident that the results are statistically significant.

(b). The authors mentioned that the disk diffusion experiments were conducted in duplicate, but they did not specify the number of independent experiments conducted, which is crucial for obtaining statistically significant results. Similarly, for the microdilution method, the number of replicates and independent experiments performed were not mentioned. It is essential to provide this information for a comprehensive understanding of the study's methodology and results.

Response: We thank the reviewer for their constructive feedback. Owing to budget constraints, each experiment was conducted once, except for the reference strains, which were evaluated in duplicate. While the limited number of replicates may affect the generalizability of the results, this study provides a basis for future large-scale research. We have added this clarification in the revised manuscript, detailing the limitations for a clearer understanding of the study's reliability (lines 352-358; 363-365).

(c). The authors do not include a conclusions section in the manuscript. Instead, they present a series of statements in the discussion section that could be interpreted as conclusions.

Response: Thank you for your valuable feedback. We agree that the manuscript would benefit from a clearer conclusion. In the revised version, we have added a separate Conclusion section, where the key findings of the study are summarized, and their implications for clinical practice are highlighted.

(d). Throughout the text, they mention that the disk diffusion and microdilution methods yield similar results, but they do not provide statistical analysis to support this claim. While there are variations in the results between the two methods, it is not specified whether these differences are statistically significant. Therefore, the assertion that the disk diffusion method provides reliable results may be premature (lines 397-398).

Response: We sincerely thank the reviewer for their valuable feedback. We refrained from making definitive conclusions on the basis of results of our study, and in addition to the validated disk diffusion method described in the literature, we have included the following statement: "However, owing to the limited sample size and potential discrepancies for certain isolates, the reliability of the disk diffusion method should be interpreted with caution, as also noted in the study". We believe that this has allowed us to present the limitations of the study more clearly.

(e). Additionally, stating that the consistent results between the two susceptibility methods validate the reliability of the disk diffusion method (lines 38-39) seems contradictory, especially when the authors themselves caution that discrepancies for certain isolates should be interpreted carefully due to the limited sample size. They suggest a larger sample is needed for future validation to minimize potential random errors. If discrepancies require further validation, so do concordances, which means that the method's validation cannot be guaranteed at this point.

Response: We would like to thank the reviewer for their insightful comments. We apologize for the definitive nature of this statement. In light of the study's limitations, we have revised the sentence to: "The results of the two antifungal susceptibility methods were compared for C. albicans and C. glabrata isolates, and the reliability of the disk diffusion test was discussed." to ensure a more cautious interpretation.

(f). The investigation of potential differences in susceptibility to topical antifungals between petite and non-petite variants of C. glabrata, as well as comparisons with other Candida species, is crucial for improving the management of vaginitis caused by these pathogens, particularly in pregnant patients for whom oral antifungals are not an option. 

Response: We appreciate the reviewer’s insight into the importance of this aspect of the study. We have expanded on the significance of examining petite and non-petite variants and their role in the management of vaginal candidiasis in pregnant women in the Conclusion section. We agree that these findings could inform more targeted treatment strategies for this population.

(g). It is also essential to conduct these tests under conditions that mimic vaginal pH to assess whether pH influences antifungal efficacy. For meaningful insights, the results must be comparable to findings from other research groups.

Response: We thank the reviewer for highlighting this important point, which we have addressed in the conclusion section, emphasizing the influence of pH on antifungal efficacy.

(h). The authors compared the disk diffusion method with the microdilution method, which is considered the reference method. However, they did not statistically analyze or extensively discuss the significant differences between the results obtained by both methods. While there are no established reference values for all antifungals in the microdilution method used in the study, variations exist even among the antifungals common to both methods. The reliability of the disk diffusion method for the analyzed samples is not fully validated based on the results presented.

Response: As indicated in the study limitations section of the manuscript, the sample size of the study is limited, and the limited sample size makes the results of antifungal susceptibility methods among microorganisms incomparable. In particular, the results of the analyses were inconclusive. Consequently, the findings are presented through numerical comparisons. If the study is replicated with a larger sample size while maintaining the rates obtained in the original study, it becomes evident that the results are statistically significant [as indicated in Response (a)].

(i). Lines 19-20: The sentence is misconstrued.

Response: We hope that the revised sentence provides a clearer interpretation: "Clinical and mycological data are essential for the optimal management of patients with Candida vaginitis (CV), particularly in cases of (i) azole-resistant C. albicans vaginitis, (ii) recurrent CV, and (iii) CV in pregnant women".

(j). Line 210: It should be "microorganisms".

Response: Done.

Reviewer 3 Report (Previous Reviewer 4)

Dear Authors

I believe that this article has improved considerably, especially with the corrections requested.

Figure 2 was much clearer in the current form.

Please revise in line 114: “the population studied (says study)

In figure 1: it was 17(%3.1) instead of 3.1%.

In line 378 they should take out ... “their” non-petite

Author Response

Reviewer 3:

  1. I believe that this article has improved considerably, especially with the corrections requested. Figure 2 was much clearer in the current form.

Response: We thank the reviewer for the feedback and are glad the revisions improved clarity.

  1. Please revise in line 114: “the population studied (says study)

Response: Thank you; this revision has been made as requested in both the text and the figure legend.

  1. In figure 1: it was 17(%3.1) instead of 3.1%.

Response: Done.

  1. In line 378 they should take out ... “their”non-petite

Response: Done.

This manuscript is a resubmission of an earlier submission. The following is a list of the peer review reports and author responses from that submission.

Round 1

Reviewer 1 Report

The author aimed to investigate the antifungal activity of six commonly used antifungals against randomly selected vaginal isolates recovered from pregnant women.

1.     The English of the manuscript is sometimes very difficult to understand. For example: line 24~26, we cannot understand which Candida species were compared. Extensive English editing of the manuscript is necessary.

2.     There are many inappropriate words. For example: azole-refractory C. albicans. Only azole-resistant is often used, and I never heard of “azole-refractory”, because the word refractory is often used in refractory septic shock or refractory cardiopulmonary failure, but not azole-refractory.

3.     The introduction is very long, please try to make it shorter. For example, line 68~72, the GOF mutations in PDR1 and upregulate efflux pumps are not related to this study, because the author did not investigate the mechanisms of antifungal resistance.

4.     In line 53~57, if the CLSI and EUCAST guidelines do not recommend the routine testing of vaginal isolates, why we should identify azole-resistant isolates and antifungal drugs that are active at vaginal PH?

5.     Line 60, suggest not to use “treatment-refractory”

6.     After finish the introduction, the readers may not be able to know why it is important to check the antifungal drugs of vaginal Candida isolates at vaginal PH of 4~4.5. Is vaginal Candidiasis very difficult to treat? Is there previous studies regarding the difficulty to treat vaginal candidiasis?

7.     There were 178 cases of CV, but only 68 isolates were selected and analyzed. Would there be selection bias? Can the 68 Candida isolates represent the whole picture of the 178 cases of CV?

8.     Overall, the result is very short, and the study only provided information of the antifungal testing of 68 isolates, may also at low PH. The case number is inadequate and all 178 isolates should be performed and analyzed, in order to have a convincing conclusion.

9.     How to identify isolates that were petite or non-petite? I found the section 2.3, However, the formations of petite and non-petite were confirmed in vitro after CV, should it be confirmed in vivo? How to know whether the specific Candida strain that form the petite or non-petite when it caused real disease? Or does it really cause petite like disease?

English writing should be revised

Reviewer 2 Report

The manuscript presents a retrospective single-centre study investigating the antifungal activity of six commonly used antifungals against vaginal isolates from 68 pregnant women in Adana, Türkiye. The isolates included various strains of Candida, and the study focused on the efficacy of antifungals at different pH levels. The study's findings emphasize the importance of topical antifungals in treating Candida vaginitis in pregnant women and the need for precise identification of Candida strains for optimal antifungal therapy.

The study addresses a critical issue in the management of Candida vaginitis, particularly in pregnant women, who represent a vulnerable group.

It provides valuable insights into the effectiveness of different antifungals, contributing to the optimization of treatment strategies.

The inclusion of various Candida species, including C. albicans, petite C. glabrata, non-petite C. glabrata, and C. krusei, offers a broad understanding of antifungal activity.

The comparison of antifungal efficacy at different pH levels adds depth to the study, considering the natural pH variations in the vaginal environment.

Highlighting the superior activity of topical antifungals (miconazole, nystatin, and clotrimazole) at specific pH levels provides practical guidance for clinicians.

The recommendation to identify petite and non-petite mutants of C. glabrata emphasizes the need for precise diagnostic methods in clinical settings.

The retrospective nature of the study may introduce biases related to data collection and patient selection.

Being a single-centre study, the generalizability of the findings to other populations or geographic regions might be limited.

The sample size of 68 pregnant women, while adequate for initial findings, may not be sufficient to draw definitive conclusions. A larger, multi-centre study would strengthen the reliability of the results.

In most hospitals their is a collection of strains. Could you collaborate and apply you methods on other strains from other centers as well? This could really offer a better understanding of the situation.

The random selection of isolates could be discussed in more detail to ensure the reader understands the selection process and its potential impact on the results.

The manuscript could benefit from a more detailed explanation of the statistical methods used, particularly how significance levels were determined and interpreted.

Graphical representations of the data, such as charts or graphs, would enhance the clarity and impact of the results.

While the discussion covers the key findings, it could be expanded to include comparisons with existing literature, highlighting similarities or discrepancies with other studies.

The conclusion should reiterate the clinical implications and potential next steps for research, offering a clear direction for future investigations.

Provide a more comprehensive description of the methodology, including isolate selection, pH adjustment procedures, and detailed statistical analyses.

Discuss the potential mechanisms underlying the observed differences in antifungal activity at different pH levels.

Incorporate graphs, charts, or tables to present the inhibition zone diameters and other key data points more effectively.

Use visual aids to highlight significant differences and trends in the data.

Provide more specific guidance on how the findings can be implemented in clinical practice, including any recommended protocols for antifungal susceptibility testing.

Discuss potential challenges or limitations in applying the study's findings to clinical settings.

Reviewer 3 Report

The study of the possible variation in susceptibility to topical antifungals between the petite and non-petite variants of C. glabrata, as well as the comparison with other Candida species, is important to ensure better management of vaginitis caused by these species, particularly in pregnant patients in whom the use of oral antifungals is contraindicated.

It is of interest that these tests are also performed under pH conditions that resemble vaginal pH, to evaluate whether there are variations in the effectiveness of the antifungal determined by pH. For this reason, the determinations must be made in a way that can be compared with the results of other research groups.

I consider that the greatest limitation of this work is limiting itself to only using the disk diffusion method for the antifungal susceptibility test when it has been seen that there are large variations, even contradictory when the susceptibility of C. glabrata is analyzed by microdilution and disk diffusion. Comparing both methods for the petite and non-petite variants would provide more information on possible differences in susceptibility to antifungals between these isolates.

Lines 24-25: It is not clear the sentence regarding the results for nystatin.

Lines 73-75: It is not clear whether yeasts are not able to ferment alternative carbon sources or are not able to ferment at all. In any case, it would be contradictory if they are not able to ferment and cannot grow on non-fermentable sources, i.e. they only grow on fermentable carbon sources. Please review this sentence.

Line 77: The word "identification" is repeated twice.

Section 2.1: Define the study period.

Lines 99-100: The sum of the Candida colonization isolates is not 17, but 16. Review and correct accordingly.

Section 2.5: Please mention the culture medium in which the antifungal tests were performed.

Lines 185-187: It is not specified at what pH the diameters were wider for nystatin. Please rewrite the sentence to make it clearer.

Section 3: Consider including some graphs that allow visualization of comparisons and significant differences between inhibition diameters.

Figure 1: Please describe the abbreviations for the antifungals on the discs.

Line 261: The abbreviation FLU-R has not been previously described.

Reviewer 4 Report

Dear Authors:

I consider this an interesting paper that presents sensitivity data to different antifungals of C. glabrata petite and non-petite isolates obtained from vaginal discharge samples of pregnant women.

You point out the value of performing sensitivity tests at vaginal pH. Unfortunately, you do not have the MIC methodology by the reference method to analyze whether the results obtained correlate well with the corresponding MIC results.

Interestingly, there is a high prevalence of Candida glabrata as an agent of vaginal candididasis in this institution in Turkey.

Why did you select 26 petite C. glabrata isolates, what were the criteria for this selection.

You could not compare the results of susceptibility with the corresponding to the reference broth method.